# Physiological Functions of Carbon Dots and Their Applications in Agriculture: A Review

**DOI:** 10.3390/nano13192684

**Published:** 2023-09-30

**Authors:** Guohui Li, Jiwei Xu, Ke Xu

**Affiliations:** 1Jiangsu Key Laboratory of Crop Genetics and Physiology/Jiangsu Key Laboratory of Crop Cultivation and Physiology, Agricultural College of Yangzhou University, Yangzhou 225009, China; lgh@yzu.edu.cn (G.L.); mz120221305@stu.yzu.edu.cn (J.X.); 2Jiangsu Co-Innovation Center for Modern Production Technology of Grain Crops, Yangzhou University, Yangzhou 225009, China; 3Research Institute of Rice Industrial Engineering Technology, Agricultural College of Yangzhou University, Yangzhou 225009, China

**Keywords:** photosynthesis, nutrient and water absorption, abiotic stress resistance, yield, biosensor

## Abstract

Carbon dots are carbon-based nanoparticles, which have the characteristics of a simple preparation process, photoluminescence, biocompatibility, an adjustable surface function, water solubility, and low-level toxicity. They are widely used in biological applications, such as imaging, biosensing, photocatalysis, and molecular transfer. They have also aroused great interest among researchers in agriculture, and there has been significant progress in improving crop growth and production. This review presents the physiological functions of carbon dots for crop growth and development, photosynthesis, water and nutrient absorption, and abiotic stress resistance and their applications in improving the ecological environment and agriculture as biosensors, and future application prospects and research directions of carbon dots in agriculture.

## 1. Introduction

The world population is constantly growing, and in order to meet the global demand for food and industrial raw materials, the input and consumption of agricultural resources continue to increase. It is estimated that the global annual crop yield is over 3 billion tons, requiring 190 million tons of fertilizers, 4 million tons of pesticides, and 2.7 trillion cubic meters of freshwater resources (accounting for 70% of global freshwater resources) [1]. Pesticides and herbicides are widely used, and low fertilizer use efficiency leads to fertilizer loss, which seriously threatens the global ecosystem and is not conducive to the sustainable development of agriculture [2,3]. The biogeosystem technique (BGT) is known to increase soil productivity and keep soil health through intrasoil milling, intrasoil pulse continually discrete watering, and intrasoil waste recycling [4,5,6]. On the other hand, aiming to address the challenges of food growth and sustainable agricultural security development, there is an urgent need for agriculture to develop new green and environmentally friendly technologies to improve crop growth and development. Nanomaterials are considered to play a crucial role in addressing these challenges faced in future agriculture. In the last few decades, a large number of nanomaterials have been developed and applied in agriculture, playing a vital role in promoting crop growth and development, increasing yield, improving fertilizer and pesticide use efficiency, protecting the ecological environment, and alleviating environmental pressure [7,8,9].

Carbon dots (CDs) are a type of carbon-based nanomaterial with fluorescence properties, consisting of carbon nanoparticles with quasi-spherical structures with a size of less than 10 nm [10]. CDs, as the generic term for a variety of nanosized fluorescent carbon materials, concretely include graphene quantum dots, carbon quantum dots, carbon nanodots, and carbonized polymer dots, which are classified according to the specific carbon core structure, surface groups, and properties, as shown in Figure 1 [11,12]. CDs generally have O/N-containing functional groups, such as amino (–NH_2_), carboxyl (–COOH), and hydroxyl (–OH), on their surface, which make these materials water soluble, probably due to the formation of H bonding [13]. CDs have abundant surface groups, which can bind to ions, organic molecules, polymers, DNA, and proteins, with the aim of changing the properties of CDs, and these functionalized CDs can be optimized to meet specific requirements in different fields. Compared with traditional nanomaterials, such as cadmium/lead, rare earths, and metal oxides, CDs have better photostability, higher quantum yield, lower toxicity levels, some from abundant low-cost sources, and excellent biocompatibility, which empowers them to have promising applications in biomedicines, optronics, sensors, and catalysis [14]. Recently, many researchers have explored the potential applications of CDs in different fields, including agriculture [15,16,17,18,19,20,21].

Different types of natural raw materials can be employed for the fabrication of CDs, and many synthetic methods have been mentioned for the fabrication of CDs in previous reports [13]. The mass production of CDs can be achieved using technologies such as electrophoresis, laser etching, and electrochemical oxidation [22,23]. Previous studies have explored the effects of carbon-based nanomaterials (such as fullerene, carbon nanotubes, and graphene particles) and metal or metal oxide nanoparticles (such as Au, Ag, Fe_3_O_4_, TiO_2_, and ZnO) on the growth of crops [24,25], demonstrating the broad application prospects of nanotechnology in agriculture. CDs are a new carbon-based nanomaterial, and research on their physiological functions and impacts on crop growth and development is still in its early stages. Since Qu et al. [26] revealed the biocompatibility of CDs with bean sprouts, the potential impact of CDs on the growth of various crops has attracted widespread attention from agricultural researchers. The application effects and mechanisms of CDs in plants, especially crops, need to be studied in the future, and new green, high-yield, and high-efficiency cultivation technologies that promote crop growth and increase yield are expected to be developed, which is of great significance for ensuring global food security.

## 2. Applications of Carbon Dots in Crop Production

Increasing agricultural productivity is critical to feeding the ever-growing human population. In recent years, the promoting effect of CDs on plant growth has been observed in monocotyledonous (wheat, rice, and maize), dicotyledonous, and other plants (mung beans, tomatoes, lettuce, tobacco, soybeans, eggplants, chili peppers, watermelons, radishes, celery, coriander, and cabbage) [10]. CDs are widely used in agriculture as seed priming agents, photosynthetic enhancers, plant stress ameliorators, and sensors [10]. The positive impact of CDs on different plants indicates their great application potential in agricultural production, which plays an important role in promoting crop growth and improving the sustainability of agricultural production. In Table 1, we summarize the growth enhancement effect of CDs on main cereal crops, rice, wheat, and maize, which account for 99% of global food production. CDs exhibit dose-dependent and mode-of-application-dependent effects on plants. CDs are introduced to the plant using various modes of application, such as seed pretreatment, foliar spraying, and hydroponic solution treatment. To fully understand the effect of CDs on crop development and the underlying mechanism, various growth parameters are evaluated throughout the full growth cycle of the plant.

## 3. The Physiological Role of Carbon Dots and Their Impact on Crop Growth and Development

The primary physiological functions of CDs are to promote seed germination and root growth, improve plant nutrient absorption, promote plant growth, increase biomass accumulation, enhance photosynthesis, and increase plant carbohydrate content, plant abiotic stress, and disease resistance parameters; these are crucial processes for the growth of plants and their crop yield (Figure 2) [2,10,25,34].

### 3.1. The Pathways of Carbon Dot Uptake by Crops and Their Accumulation and Transport Characteristics within Plants

The main pathways for CDs to enter plants include root absorption from soil/water and leaf absorption. Studies have shown that CDs can penetrate plant cells, and then be transported along with water and minerals from the roots to the stems and leaves; they are absorbed through cell walls and plasmodesmata via extracellular pathways in intercellular spaces and extracellular spaces, and then pass through the cortex and enter the xylem through the plastid pathway [35]. CDs have stable and unique fluorescence signals, providing a good pathway for tracking in plants. The process of CDs being absorbed by plants can be demonstrated using methods such as fluorescence imaging, transmission electron microscopy, or Raman spectroscopy measurement [25,30,36,37,38]. Using a CD aqueous solution to cultivate mung beans, concentration-dependent red orange fluorescence enhancement can be clearly observed in mung bean seedlings under 365 nm ultraviolet light. The roots, stems, and leaves of young seedlings were observed using confocal laser scanning microscopy, and it was found that CDs mainly exist in the vascular system. Transmission electron microscopy was used to observe the transverse sections of roots, stems, and leaves, and large aggregation clusters of CDs were observed in the intercellular spaces. Therefore, CDs are absorbed by the root and enter the root vascular bundle, which is then transported to the stem and leaf vascular bundles, and then enter the intercellular space for aggregation [39]. In addition, studies on crops such as rice and corn have shown that CDs can be absorbed and used by plants through foliar spraying, thereby increasing the grain weight and yield [25,31].

### 3.2. Carbon Dots Enhance Crop Photosynthesis

The growth and yield of crops depend on effective photosynthesis, which is the fundamental material source for plant growth and biomass accumulation, contributing over 90% of crop biomass. One important physiological function of CDs is to enhance plant photosynthesis. Photosynthesis includes two energy conversion processes from light energy to electric energy and from electric energy to chemical energy, involving light absorption, electron transfer, photophosphorylation, carbon assimilation, and other important reaction steps. CDs typically exhibit strong absorption in the ultraviolet region (200–400 nm), but their light absorption can be extended to the visible light range due to the type and content of surface groups, as well as changes in the oxygen/nitrogen content in carbon nuclei. In the 500–800 nm range, CDs exhibit longwave absorption, converting UV light that cannot be used by plants into visible light [18,40,41]. CDs are both excellent electron donors and electron acceptors [31,42,43]. Amine functionalized CDs are strongly conjugated on the surface of chloroplasts and assist in absorbing photons to transfer electrons to the chloroplasts, accelerating the all-electron transfer chain pathway in photosynthetic reactions, thereby enhancing photosynthesis [44]. Five mg L^−1^ nitrogen-doped CDs significantly increased the net photosynthetic rate of maize (21.51%). Further studies have shown that there is an increase in the light conversion rate, electron supply, chlorophyll content, ATP synthase (adenosine triphosphate synthase) activity, and NADPH (nicotinamide adenine dinucleotide phosphate) synthesis during photosynthesis, with a significant increase of 122.80% in the electron transfer chain rate [31,33]. In addition, CDs significantly increase the expression of chlorophyll synthase and chlorophyll enzyme genes in rice, which helps to improve chlorophyll synthesis and CO_2_ assimilation [27,29]. CO_2_ assimilation is a physiological process responsible for the conversion of electrical energy to chemical energy in photosynthesis, and rubisco is a key enzyme in this process [45]. Rubisco enzyme activity directly affects the photosynthetic rate and carbohydrate accumulation. Wang et al. [37] found that the rubisco enzyme activity of mung bean seedlings treated with CDs was 30.9% higher than that of the control. Similar increased effects of CDs on rubisco enzyme activity were found in plants such as rice and arabidopsis [25,46]. In terms of enhancing photosynthesis, CDs have more advantages in monocotyledonous plants than dicotyledonous plants do. The structural differences in the vascular system and root structure of monocotyledonous and dicotyledonous plants are the reasons for the excessive photosynthesis [10]. CDs also have different effects on the photosynthesis of C3 plants (rice) and C4 plants (corn), and their effect on the CO_2_ assimilation of rice is larger than that of corn, which is because corn is a C4 plant that has an internal way to reduce photorespiration and improve the CO_2_ fixation rate; CDs also significantly improve the stomatal conductance of rice. The greater the stomatal conductance is, the higher the CO_2_ absorption rate in the stomata is; therefore, CDs can enhance the gas exchange capacity of plant leaves [28,47]. In addition, CDs can be degraded in plants to form plant hormone analogues and release CO_2_ and hormone analogues to promote plant growth, and CO_2_ released is further assimilated through the Calvin cycle, thus increasing carbohydrate accumulation [25,46,48]. The above research indicates that CDs have great potential to improve crop growth and photosynthesis in agricultural production. In conclusion, CDs provide artificial photosynthesis support for crops, which increases the photosynthesis rate and, consequently, increases the grain yield. The morphological characteristics of plants, such as plant height, biomass, and leaf area, and the physiological characteristics, such as stomata conductance, rubisco activity, ATP and NADP formation, PSI and PSII rates, and electron transfer chain, are all improved by CDs [49], which raises the carbohydrate levels and, finally, increases grain yield.

### 3.3. Improving Crop Quality with Carbon Dots

Research has shown that CDs can be applied as nanofertilizers, improving crop photosynthesis, while also increasing crop yield. Spraying 560 mg L^−1^ of CD aqueous solution on leaves can increase the yield of dicotyledonous plants, such as soybeans, tomatoes, and eggplants, by 20% [46]. Wang et al. [37] found that 0.02 mg mL^−1^ CDs increased the root length, stem length, root activity, and fresh weight of mung beans, resulting in a 17.5% increase in bean sprout yield compared with that of the control.

Photosynthesis is the main source of assimilation for crop yield formation, and the level of yield is determined by the accumulation and distribution of photosynthetic products. The continuous spraying of CDs (50 mg L^−1^) during the vegetative growth stage of maize can increase carbohydrate accumulation during the reproductive growth stage, resulting in an increase in the 1000-grain weight and a final yield increase of 24.50%. One possible reason is that the expression of sucrose transporter (SUT) genes in leaves increased 1.61 times after treating with CDs, and the upregulation of SUT expression enhanced the transportation capacity of sucrose in the phloem. Therefore, more carbohydrates are transported from leaves to grains, thereby promoting grain filling and increasing yield [31]. Under drought conditions, spraying nitrogen-doped CDs on maize leaves promotes the synthesis of carbohydrates by enhancing photosynthesis, resulting in a 30% reduction of maize yield loss; at the same time, the starch, soluble sugar, protein, linoleic acid, and α-linolenic acid contents of grains significantly increased by 7.0%, 9.8%, 49.7%, 10.5%, and 12.3%, respectively, thus improving the grain quality [33]. An appropriate concentration of nitrogen-doped CDs can significantly promote the accumulation of lettuce biomass, significantly improving the soluble sugar and other nutritional quality indicators of lettuce [50].

Fertilizer, as the main source of crop nutrients in modern agricultural production, directly participates in or regulates crop nutrient metabolism and cycling and is closely related to crop yield and quality; adding CDs as fertilizer enhancers to different types of fertilizers can accelerate chemical reactions, accelerate nutrient decomposition, improve fertilizer release characteristics, increase fertilizer use efficiency, and promote crop growth and development [51]. Adding nanocarbon to slow-release fertilizers can promote the formation of rice tillers, increase the chlorophyll content during booting, promote dry matter accumulation, and increase the number of effective panicles and grains per panicle, ultimately leading to an increased grain yield and nitrogen fertilizer use efficiency in rice [52]. Therefore, CDs have great potential in improving crop yield and quality.

### 3.4. Carbon Dots Promote Seed Germination and Increase Water and Nutrient Absorption

Seed germination is the first and most crucial step in plant growth, and good seed germination can help plants develop better. It has been found for both rice and wheat that seeds treated with a CD aqueous solution can promote seed germination [25,26]. Water and nutrient absorption and assimilation are important factors affecting seed germination, crop growth and development, and yield formation. One of the reasons that CDs promote seed germination and enhance seed vitality is that they can penetrate the hard seed coat, promote water infiltration, and facilitate seed water absorption and germination [10]. Seed germination, root development, seed moisture content, and seedling length are related to the surface hydrophilic groups (–OH and –COOH) of CDs. The hydrophilic groups on the surface of CDs provide rich binding sites for water molecules, and they are absorbed by the plant; adequate water absorption promotes seed germination and accelerates seedling growth [25,46]. In addition to serving as adsorption sites for water, CDs upregulate the expression of seed aquaporin genes, activate aquaporins, and reduce the rhizosphere pH, which promotes water and nutrient absorption, improves the rhizosphere microbial environment, and is conducive to seed germination and growth [53,54].

The absorption of water by crops is accompanied by the absorption of nutrients. Hydroxyl and carboxyl groups also endow CDs with the ability to adsorb various nutrient ions (K^+^, Ca^2+^, Mg^2+^, Cu^2+^, Zn^2+^, Mn^2+^, and Fe^3+^), which are important nutrient elements for crop growth. They interact with hydrophilic groups on the surface of CDs through hydrogen bonding and electrostatic interactions, and adsorb on the surface of CDs; when CDs enter the plant, the concentration of nutrient ions increases with CDs in the plant, and this is also the reason for the sustained and slow release of nutrients from the xylem [55]. The nutrient ion content in arabidopsis treated with CDs is higher than that in a control, indicating that nutrient ions can enter the plant with CDs [46]. Studies on coriander showed that spraying 40 mg L^−1^ of CDs increased the content of K, Ca, Mg, P, Mn, and Fe by 64.3%, 21.0%, 26.2%, 12.8%, 56.0%, and 125%, respectively [56]. When 0.02 mg mL^−1^ of CDs was used to treat lettuce, the N, P, and K contents of the plants increased by 4.4%, 10.8%, and 16.5%, respectively [57]. On the other hand, after treating them with CDs, the expression of genes related to aquaporins (membrane proteins used for transporting water, nutrients, and gases) is upregulated, thereby increasing water and mineral ion absorption and promoting crop growth [58,59]. Carbon is widely present in soil, and CDs are more environmentally friendly than general nanomaterials are. The CDs entering the soil increase the EC value of soil. The increase in EC value is the direct reason for the formation of a large amount of bicarbonate ions in soil, promoting the absorption and utilization of water and nutrients, such as nitrogen, phosphorus, and potassium, via crop roots [60]. In addition, CDs can improve the activity of nitrogenase and the nitrogen fixation efficiency of nitrogen-fixing bacteria by affecting the secondary structure of nitrogenase and improving electron transfer in the biocatalytic process. This provides an economic and environmentally friendly way to improve the biological nitrogen fixation ability of nitrogen-fixing bacteria and provide a nitrogen source for crop growth when nitrogen is insufficient [2,37].

### 3.5. Carbon Dots Improve Crop Resistance to Abiotic Stress and Disease Resistance

In actual production, crops are constantly challenged by adverse abiotic environmental conditions. According to a report of the Food and Agriculture Organization of the United Nations (FAO), in 2007, more than 96% of the world’s rural land was affected by various abiotic stresses, including droughts, high temperatures, low temperatures, nutrient deficiency, and excessive salt or heavy metals in the soil. These abiotic stresses have a negative impact on crop productivity, leading to serious yield losses [61,62]. For example, drought stresses on rice and wheat after flowering can lead to premature plant senescence, reduced material production, a shortened filling period, and a reduced grain weight [63,64,65]. CDs can promote plant growth, and a large number of studies have also proved that they can improve plant resistance to abiotic stress, thereby improving crop yield.

The increase in reactive oxygen species (ROS) is the main factor affecting crop growth under abiotic stress. The accumulation of ROS in cells usually leads to oxidative damage to proteins, lipids, carbohydrates, and DNA [66,67]. The mechanism of CDs improving crop resistance to abiotic stress is that CDs have the effect of scavenging free radicals, and the surface of CDs fused with carboxyl and amino groups can transform DPPH free radicals into stable DPPH-H through a hydrogen transfer mechanism [68,69]. On the other hand, CDs can increase the activity levels of superoxide dismutase (SOD), peroxidase (POD), and catalase (CAT) and reduce the contents of ROS and malondialdehyde (MDA) [70]. CDs combine the free radical scavenging characteristics and the ability to improve the activity of antioxidant enzymes to protect crops from abiotic stress, providing a theoretical basis for their application in crop-stress-resistant cultivation. Research has shown that under a drought stress, the application of CDs in maize reduces the accumulation of ROS within the plant, weakens the oxidative stress caused by the drought, and promotes the synthesis of proline and abscisic acid in the leaves and long-distance transportation to the roots, thereby upregulating the expression of aquaporin (*AQP*) genes from 2.3 to 7.6 times. This increases the proportion of K^+^/Na^+^ (by 47.7%) and promotes crop water absorption under drought conditions (by 49.0%); in addition, it increases the transport rate of carbohydrates to the roots, increases the content of root exudates (organic acids, amino acids, etc.), and increases the abundance of rhizosphere microorganisms (proteobacteria, actinomyces, ascomycetes, etc.), promoting the absorption of nitrogen and phosphorus in maize, thereby improving the drought resistance of these crops [33]. CDs also have a positive impact on the growth of soybeans under drought conditions; applying 5 mg L^−1^ CDs on the leaves can eliminate the accumulation of reactive oxygen species in soybean leaves under a drought stress, enhancing photosynthesis and carbohydrate transport. On the other hand, CDs stimulate the root to secrete amino acids, organic acids, auxin, and other organic substances; recruit beneficial microorganisms for the rhizosphere (such as actinomyces, ascomycetes, acidobacteria, and mycobacterium). and promote the activation of soil nitrogen. At the same time, the expression of the *GmNRT*, *GmAMT*, and *GmAQP* genes in the root was upregulated, promoting nitrogen absorption, utilization, and metabolism, significantly improving the nitrogen content and water absorption capacity of soybeans. In addition, it also increased the protein, fatty acid, and amino acid contents in soybean seeds by 3.4%, 6.9%, and 17.3%, respectively [71,72]. Wang et al. [73] explored the relieving effect of CDs on heat stress in Italian lettuce. The study showed that CDs enhanced the antioxidant enzyme activity and osmoregulation (manifested by a decrease in proline content) and reduced the damage caused by lipid peroxidation in plant cells (manifested by a decrease in MDA levels), thereby improving the plant’s heat tolerance. Soil salt stress is also an important factor affecting plant growth. Salt stress leads to crop nutrient deficiency, osmotic stress, and ion toxicity. Li et al. [74] showed that CDs exhibit Ca^2+^ mobilization characteristics and can alleviate salinity stress by enhancing Ca^2+^ signaling and ROS scavenging activity. In addition, Gohari et al. [75] found that putrescine-functionalized CDs can increase the K, photosynthetic pigment, proline, and phenolic substance contents and antioxidant enzyme activity levels in grape leaves, while reducing the Na^+^, MDA, and H_2_O_2_ contents, thereby alleviating salt stress and increasing the leaf fresh and dry weights. The above research provides a theoretical basis for CDs as abiotic stress modifiers to improve crop yield and plant protection.

Crop diseases are important limiting factors that affect crop growth and yield formation. Currently, prevention and control measures mainly rely on the widespread use of pesticides/fungicides, and the inefficient use of pesticides seriously threatens ecosystems’ biodiversity and functions. A previous study found that CDs have broad-spectrum antibacterial activity against bacteria and fungi, laying the foundation for their application in crop disease control and improving crop disease resistance [76]. CDs can destroy the secondary structure of DNA/RNA in bacterial walls and bacteria and fungi, thus showing broad-spectrum antibacterial/antifungal activity against Gram-positive (*Staphylococcus aureus* and *Bacillus subtilis*) and Gram-negative (*Bacillus* sp. *WL-6* and *Escherichia coli*) bacteria [2,25]. Luo et al. [77] found that foliar spraying 10 mg L^−1^ nitrogen-doped CDs activated the acquired resistance dependent on jasmonic acid—and salicylic acid—in tomatoes, resulting in the stagnation of pathogen growth in vivo, effectively reducing the symptom severity of tomato green wilt syndrome caused by Ralstonia solanacearum by 71.19%. Therefore, CDs can be used as green and efficient antibacterial agents for preventing and controlling crop diseases.

### 3.6. The Protective Effect of Carbon Dots on Agricultural Ecological Environment

A good ecological environment is the key to food security production. Given the low toxicity, chemical inertness, biodegradability, and low cost of CDs, they are considered an excellent choice for degrading organic pollutants. Pesticides, industrial chemicals, by-products of industrial processes, and aromatic hydrocarbons entering the environment can induce and accumulate pathogenic bacteria in the food chain, transforming them into organic pollutants. These pollutants have mutagenic, carcinogenic, and endocrine-disrupting chemical properties, which harm human health [13,78]. The existing research has recognized the key role of CDs in the photodegradation of organic pollutants, such as residual pesticides and endocrine disruptors. Cruz et al. [79] found that CDs were used in combination with a photosensitizer for the photodegradation of organic dyes and toxic gases. Compared with their original photocatalysts, the addition of CDs greatly improved the degradation rate. CDs have the potential to be used as photocatalysts for tetracycline degradation. Tetracycline is a broad-spectrum antibiotic, which can promote the growth of livestock and poultry at low doses and can be used to treat animal diseases at high doses. Because of its widespread use in the livestock and poultry and aquaculture industries, it releases certain residues in the environment, and it poses a more prominent environmental problem. Recent research shows that, without any additives, nitrogen-doped CDs can completely degrade tetracycline within 10 min, thereby eliminating its adverse effects on the ecological environment [80]. In addition, CDs can also reduce the absorption of heavy metal ions by crops, thereby avoiding the harm caused by heavy metals to crop growth [81], because heavy metal ions can be trapped between functional groups on the surface of CDs, thus reducing the concentration of heavy metal ions in crops [82]. In addition, CDs have a certain activation effect on soil urease, dehydrogenase, and catalase, changing the soil pH, increasing the soil microbial community abundance and diversity, increasing Cd^2+^ fixation, reducing Cd^2+^ mobility, and thus reducing the harm caused by Cd^2+^ on the environment [83]. Safe water is a fundamental need that promotes healthy crop production. However, the massive and continuously increasing consumption of hazardous chemicals at the domestic and industrial levels has led to a large amount of wastewater discharge and ecological problems. Green, synthesized carbon-dot-based photocatalysts for wastewater treatment have the advantages of being inexpensive and nontoxic, require basic ingredients, exhibit quick reactions, and involve easy procedures and straightforward postprocessing steps [13,78,84]. Our understanding of the carbon-dot-based photocatalytic degradation of contaminated wastewater will also set the stage for future wastewater treatment research.

### 3.7. The Role of Carbon Dots as Biosensors in Agriculture

Due to the fluorescence-quenching phenomenon, as the concentration of pesticides increases, the fluorescence intensity of CDs significantly decreases. Based on this, Tafreshi et al. [85] developed a fluorescence-based method for the detection of three pesticides, including diazinon, glyphosate, and semicarbazide. This sensor has good selectivity even in the presence of other herbicides and demonstrated an excellent performance using tomato samples. It can also be used in other agricultural products to achieve the ultrasensitive fluorescence detection of pesticides in real samples using green CDs. When electrostatic interactions occur between functional groups on the surfaces of CDs and isothio pesticides, CDs exhibit fluorescence quenching, and Ghosh et al. [86] developed a sensor for the detection of isothio pesticides based on the fluorescence-quenching phenomenon of CDs. The detection limit is as low as nanomolar concentrations in water, as well as in fruit and rice samples, but it is still highly selective in the presence of various interfering substances and other pesticides. The fluorescence properties of CDs can be significantly regulated through atomic doping or surface functionalization. Doped CDs can be single-element-doped or multi-element-doped, which enhances the fluorescence characteristics of CDs and significantly improves their performance in sensing applications [87,88]. Yahyai et al. [89] synthesized sulfur- and nitrogen-doped CDs, and developed a paper-based a chemiluminescence sensor using these CDs for the determination of oxfamil pesticide in fruit juice and water. It was found that KMnO_4_ oxidized S and N-CDs and produced their excited state, which significantly enhanced the luminous intensity of CDs. Therefore, the sensitivity and detection capability of the sensor were significantly improved. In addition, the sensor showed a good recovery rate between 97.5% and 105.5% for actual samples, which indicates the good performance of the sensor. Luo et al. [90] developed B, N-doped CDs and fluorescence platforms based on cerium oxide (CeO_2_), and the principle was that CeO_2_ performed the catalytic reaction of p-methyl-p-oxyphosphorus in an alkaline medium, converting it into p-nitrophenol and phosphate, reducing the fluorescence activity of B, N-CDs, which can be used to detect the residues of methyl-p-oxyphosphorus pesticides in herbal plants and water. CDs can also be used for wastewater remediation via the sensing of heavy metal ions (e.g., Pb^2+^, Hg^2+^, Cr^6+^, Cd^2+^, or Cu^2+^) and various anions in water bodies [12], which offer a feasible strategy for safe crop production and the protection of human health and the environment.

### 3.8. Differences between Application of Carbon Dots and Traditional Technologies on Crop Production

Through the above review, we compared the differences between carbon dots and traditional cultivation techniques in agricultural applications, mainly manifested in the following aspects. First, carbon dots can improve crop nutrient absorption and utilization efficiency, promote crop growth and development, and thereby increase crop yield by regulating plant physiological processes and metabolic pathways [2]. Traditional technologies mainly increase crop yield by improving soil fertility, fertilization, and pest control. Second, CDs functionalized with different functional groups can serve as carriers to accurately control and release agricultural inputs, such as pesticides, fertilizers, and water; improve the utilization efficiency of pesticides and fertilizers; reduce dosage; and reduce waste and environmental pollution [10,11]. However, traditional technologies are relatively extensive in the use of agricultural input materials, which can easily lead to resource waste and environmental pollution. Third, carbon dots can reduce the impact of environmental stress on crops by enhancing their resistance to stress and pests [2,25]. In contrast, traditional technologies mainly rely on chemical pesticides and biological control methods for disease and pest control, with relatively limited effectiveness. Fourth, carbon dots reduce the use of pesticides and fertilizers in crop production, reducing environmental pollution while reducing agricultural production costs. CDs come from abundant sources, have low costs, and are low in toxicity to the environment [14]. In contrast, the extensive use of pesticides and fertilizers in traditional technologies requires a significant amount of energy consumption and high costs, and low fertilizer utilization rates can easily cause environmental pollution, reducing agricultural economic benefits. In summary, the application of carbon dots has advantages in crop production compared with traditional technologies. However, the application of CDs also requires relevant environmental risk assessment and management [10,36]. Further strengthening the application research of carbon dots in agriculture in the future is of great significance for sustainable agricultural development.

## 4. Futuristic Outlook

Although considerable developments have been made in the research on CDs, further research is needed to broaden the understanding of their potential application in agriculture. Here, future research prospects of CDs in agriculture are discussed in brief (Figure 3).

Farmland in China with low and middle yields accounts for 90% of the total arable land in the country. Improving the crop yields of these types of farmland is an important issue that urgently needs to be solved for food production in China, and an important way to implement the strategy of “storing grain in the land” and ensure the food security of China [91]. The traditional methods of improving soil is to increase the application of nitrogen, phosphorus, potassium, organic fertilizers, biochar, etc. These methods require fossil fuel consumption, and the cost and energy consumption are high. In addition, the soil water and fertilizer retention capacities are generally poor in medium- and low-yield fields, which are prone to nutrient loss and environmental pollution and are not conducive to the sustainable development of agricultural production, making it difficult to promote large-scale production. Given the low cost, nontoxicity, and excellent physiological effects of CDs, it is of great significance to regulate them from the perspective of the crops themselves. In the future, we will conduct research on the impact of CDs on the growth, development, and yield formation of different crops in medium- and low-yield fields and develop new environmentally friendly cultivation and regulation technologies to improve the production capacity of medium- and low-yield fields in China.

The nonstructural carbohydrates stored in the stems and sheaths of cereal crops (such as rice, wheat, and corn) are important sources of grain filling substances, and their transport rate affects the grain filling rate, thereby affecting the grain weight and yield [92,93]. Vascular bundle development, hormone levels, carbon metabolism enzyme activity, and sucrose transporter, SWEET gene expression, and protein levels are important factors affecting photosynthetic transport and grain filling [94,95,96,97]. CDs were found to upregulate the expression level of sucrose transport protein genes in maize, which may enhance the sucrose transport ability of the phloem, thereby increasing the transport of photosynthetic assimilates to grains, promoting grain filling and yield formation [25,31]. Therefore, future research on the regulatory role and effect of CDs on stems has important guiding significance for the green, high-yield, efficient, and high-quality cultivation of rice.

Crop growth conditions, including temperature, soil, water, and cultivation management, have a significant impact on crop growth and yield formation. There is currently a lack of research on whether they will affect the application of CDs in agriculture. Temperature has a significant impact on the biological activity and metabolic processes of crops. Appropriate temperature can promote crop growth and development, improve photosynthesis efficiency, and increase nutrient absorption and utilization efficiency. At the same time, temperature can also affect the activity of microorganisms in the soil, and thereby may affect the degradation and utilization of carbon quantum dots. The properties of soil have a direct impact on crop growth. Different types of soil have different fertility and water retention abilities, which can affect crop growth and yield. Under field conditions, it remains to be determined whether factors such as soil pH, organic matter content, and microbial activity will affect the stability and effectiveness of CD uptake by crops through their roots. Water is one of the key factors for crop growth. Appropriate water conditions can promote crop growth and nutrient absorption, thereby increasing crop yield. Future research needs to focus on the impact of soil moisture conditions on the dispersion and stability of CDs, as well as how to improve their utilization efficiency in soil. Reasonable cultivation and management measures can optimize the growth environment of crops and improve their adaptability and yield. For example, measures such as timely fertilization, pest control, and appropriate cultivation density can improve the nutrient absorption and utilization efficiency of crops, thereby promoting crop growth and yield formation. The effectiveness of cultivation management measures on the application of CDs in crops still needs to be explored. Therefore, future research needs to comprehensively consider the interaction between environmental conditions, cultivation management measures, and CDs on crop growth, development, and yield formation in order to optimize CD application technology, maximize CD advantages, and improve crop productivity.

Previous studies have confirmed that CDs have antibacterial effects and can enhance plant abilities to resist plant pathogens and pests, but they may also pose a potential threat to beneficial microbial communities. Therefore, it is necessary to evaluate their potential impact on nontarget organisms to ensure their safe application to agriculture. In addition, the impact of CDs on crops depends on the application dosage, and it varies from crop to crop. Numerous studies have reported the benefit of CDs on crop growth. However, high doses of CDs can also have adverse effects on plant growth, such as those from 1000 to 2000 mg L^−1^ CDs, which significantly reduced the fresh weight of roots and aboveground parts of seedlings in maize [36]. CDs cause a decrease in the growth rate of microalgae in a suspension, resulting in a decrease in available light flux due to their absorption of light (shading effect) [98]. Therefore, dosage optimization is necessary for different crops before applying CDs in order to improve their application effectiveness in large-scale production. In addition, the existing research lacks a sufficient understanding of the bioaccumulation of CDs in animals and plants and their subsequent cumulative effects on the food chain. Therefore, conducting systematic and in-depth research in this field will elucidate the ecological toxicity of CDs and contribute to their safe application to agriculture.

Sensors based on CDs have great prospects in agricultural monitoring, but the research in this field is still in its early stages. The previously reported CD-based sensors are excellent candidates for detecting various pesticide residues in agricultural products, including insecticides and herbicides. In this field, future research is still needed to accelerate the construction of a highly sensitive sensor platform for the rapid sensing of food and other products, and the development of an integrated, convenient, and wearable sensor is needed to achieve the real-time and on-site detection of toxic chemicals and heavy metals and the monitoring of plant growth, micronutrients, and abiotic and biotic stresses on plants.

In addition, the easy surface functionalization of CDs indicates that they can serve as promising carriers to deliver specific molecules to crops, such as insecticides, nutrients, and plant growth regulators, thereby achieving the precise regulation of crop growth and development.

## Figures and Tables

**Figure 1 nanomaterials-13-02684-f001:**
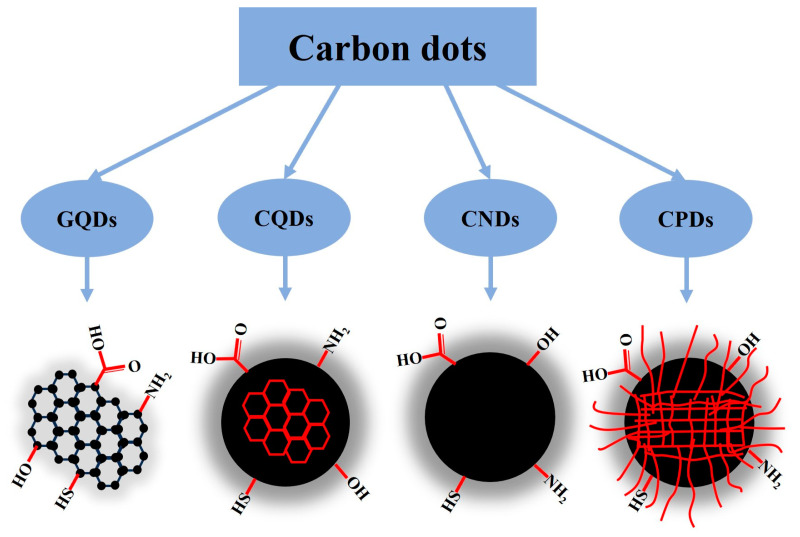
Classification of carbon dots: GQDs, or graphene quantum dots, are small graphene fragments consisting of a single graphene sheet or a few graphene sheets with obvious graphene lattices and chemical groups on the edge or within the interlayer defect; CQDs, or carbon quantum dots, are always spherical and possess obvious crystal lattices and chemical groups on the surface; CNDs, or carbon nanodots, possess a high carbonization degree with some chemical groups on the surface, but usually show no obvious crystal lattice structure and polymer features; CPDs, or carbonized polymer dots, possess a polymer/carbon hybrid structure comprising abundant functional groups/polymer chains on the surface and a carbon core [11,12].

**Figure 2 nanomaterials-13-02684-f002:**
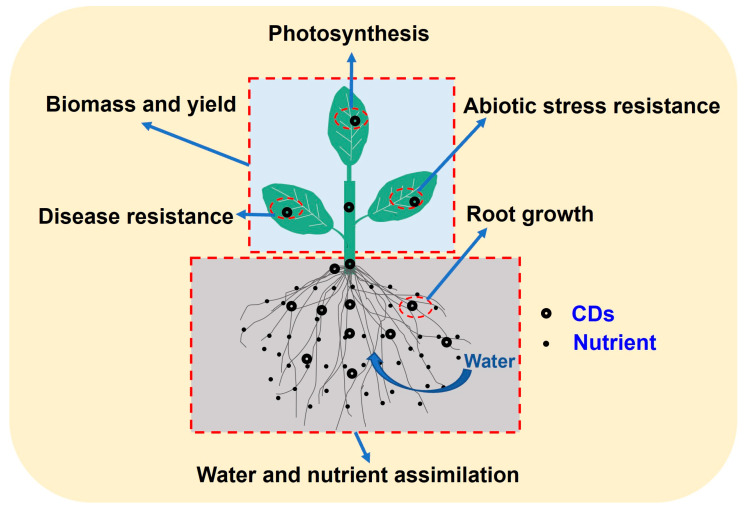
Physiological functions of carbon dots.

**Figure 3 nanomaterials-13-02684-f003:**
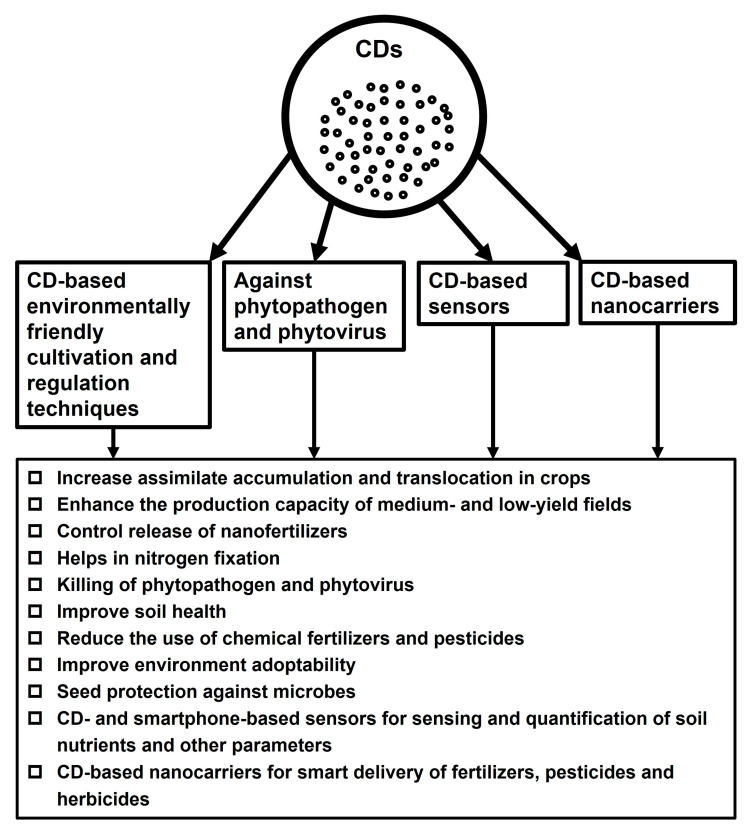
Future aspects of applications of CDs in modern agriculture.

**Table 1 nanomaterials-13-02684-t001:** Effect of CDs on plant growth of main cereal crops.

Crop	Source of CDs	Treated Part of Plant	Experimental Methods	Effects of CDs on Plant	Mechanism of Action	Reference
Rice	Electrochemical etching using graphite rods; CDs are about 5 mm	Seed	Soak in MS medium supplemented with CDs of 0.56 mg/mL and expose for 10 days	Enhance disease resistance ability and grain yield	Increase the thionin gene expression; CDs degrade to form Plant hormone analogues and CO_2_; RuBisCO activity increases by 42%	[25]
Rice	Microwave pyrolysis using citric acid and ethanolamine; CDs are 3–4 nm	Leaf	Spray with 300 μg/mL CDs 3 times a week with 5 mL/pot	Increase shoot length and dry weights of shoot and root	Increase the electron transport rate and photosynthetic efficiency of photosystem II by 29.81% and 29.88%, respectively; increase the chlorophyll content and RuBisCO carboxylase activity by 64.53% and 23.39%, respectively	[27]
Rice	Microwave pyrolysis using biochar; CDs are 1–4 nm	Leaf	Spray with 150 mg/mL CDs twice a month until the end of heading	Increase plant height and grain weight by 4.8% and 5.1%, respectively	Increase CO_2_ assimilate rate by 56%	[28]
Rice	Hydrothermal using citric acid, ethanolamine, and magnesium hydrate	Leaf	Spray with 50, 100, and 300 μg/mL CDs at a dosage of 5 mL/pot and exposure for 16 days	Increase the height and fresh biomass by 22.34% and 70.60%, respectively	Upregulate the gene expressions of enzymes related to chlorophyll by 15.26–115.02%; increase chlorophyll a and chlorophyll b contents by 14.39% and 26.54%, respectively; increase the RuBisCO activity plants by 46.62%	[29]
Wheat	Carbon soot of mustard oil lamp; CDs are 20–100 nm	Seed	Soak in CD aqueous solution for 3–4 days	Increase root and shoot lengths	Regulate the movement of water and ions	[30]
Maize	Thermopolymerization of melamine and ethylene diamine tetraacetic acid; CDs are 2.5 nm	Leaf	Spray 5 mL per plant with 1, 5, 10, and 50 mg/L CDs and expose for 7 days	Increase yield and 1000-grain weight by 24.5% and 15.0%, respectively	Increase light conversion efficiency by 121.00%; increase chlorophyll content by 15.41%; increase relative gene expression of psbA by 22.30-fold, ATPase activity by 41.44%, and NADPH production by 110.31%	[31]
Maize	Microwave pyrolysis using biochar; CDs are 1–4 nm	Leaf	Spray with 150 mg/mL twice a month until the end of heading	Increase plant height and ear weight by 20.9% and 39.6%, respectively	Increase CO_2_ assimilate rate and stomatal conductance by 16% and 18%, respectively	[28]
Maize	Ultrasonication using citric acid and *o*-phenylenediamine	Root; leaf	Hydroponic medium cultivation and spraying on leaves with 1, 5, and 10 mg/L CDs and expose for 14 days	Increase photosynthetic parameters	Increase photosynthetic pigments	[32]
Maize	Hydrothermal using citric acid and ethylenediamine	Leave	Spray 5 mL per plant using 5 mg/L CDs solutions, and expose for 33 days	Increase fresh weight of shoots and roots by 62.1% and 50.6%, respectively, and dry weight of shoots and roots by 29.2% and 37.5%, respectively	Increase net photosynthesis by 122.9%; increase root exudates of succinic acid (14.5 folds), pyruvic acid (10.0 folds), and betaine (11.8 folds), and relative abundance of microbial community by 122.6–344.4%	[33]

## Data Availability

No new data were created or analyzed in this study. Data sharing is not applicable to this article.

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
