# Peer review of "Physiological Functions of Carbon Dots and Their Applications in Agriculture: A Review"

_nanomaterials, 2023, doi:10.3390/nano13192684_

Round 1
Reviewer 1 Report
Some relevant aspects on physiological functions of carbon dots and their applications for plant growth and development are presented in the paper.
The study is interesting, the subject is topical in the related field, but the text should be very carefully revised and many sentences reformulated. I made some suggestions in the attached document.

The quality of English language should be improved.
Author Response
Response to Reviewer 1 Comments
Point 1: Some relevant aspects on physiological functions of carbon dots and their applications for plant growth and development are presented in the paper.
The study is interesting, the subject is topical in the related field, but the text should be very carefully revised and many sentences reformulated. I made some suggestions in the attached document.
Response: Thanks for your comments and suggestions. We carefully revised the manuscript accoroding to your suggestions in the attached document.
Point 2: The quality of English language should be improved.
Response: The manuscript has undergone English language editing by MDPI. The text has been checked for correct use of grammar and common technical terms.

Reviewer 2 Report
Dear Authors,
Your manuscript is a valuable scientific contribution. Physiological functions of carbon dots are an important issue.
I propose you to give explanations and make corrections. The details are presented herewith below.
Major comments
The manuscript is a review. Please indicate this in the title.
Lines 26-27
…requiring 190 million tons of fertilizers, 4 million tons of pesticides, and 2.7 trillion cubic 26 meters of freshwater resources (accounting for 70% of global freshwater resources).
You are right. However, most of fertilizers, pesticides and water are wasted. We have proposed a Biogeosystem Technique methodology (https://doi.org/10.1021/acsomega.0c02014; https://doi.org/10.1021/acsomega.0c04906; doi: 10.1007/s10653-023-01550-7; https://doi.org/10.3390/agronomy12112765) to overcome the standard technology flaws. It should be useful to reflect this in the Introduction section.
You have focused your manuscript exclusively on the carbon dots.
At the same time, the growth conditions – soil, water regime, crop production management – are important. There is a strong need to discuss this along with the carbon dots prospects in agronomy.
Minor comments
In the agronomy studies, it is common to compare new and old technology. You have failed to provide this in the Table 1 and in the text as well.
P.S. Please keep in mind that our publications are listed not for a mandatory citation.
Author Response
Response to Reviewer 2 Comments
Point 1: The manuscript is a review. Please indicate this in the title.
Response: Thanks. The title has been revised to “Physiological Functions of Carbon Dots and Their Applications in Agriculture: A review”
Point 2: Lines 26-27…requiring 190 million tons of fertilizers, 4 million tons of pesticides, and 2.7 trillion cubic 26 meters of freshwater resources (accounting for 70% of global freshwater resources).
You are right. However, most of fertilizers, pesticides and water are wasted. We have proposed a Biogeosystem Technique methodology (https://doi.org/10.1021/acsomega.0c02014; https://doi.org/10.1021/acsomega.0c04906; doi: 10.1007/s10653-023-01550-7; https://doi.org/10.3390/agronomy12112765) to overcome the standard technology flaws. It should be useful to reflect this in the Introduction section.
Response: Thanks. We have read these articles and added relevant content in the introduction of the revised manuscript.
Point 3: You have focused your manuscript exclusively on the carbon dots.
At the same time, the growth conditions – soil, water regime, crop production management – are important. There is a strong need to discuss this along with the carbon dots prospects in agronomy.
Response: Thanks. In the revised manuscript, we discussed the necessity of strengthening research on the effects of the interactions of crop growth conditions such as temperature, soil, water and cultivation management with carbon dots on crop growth and yield formation in the section of “4. Futuristic outlook”.
Point 4: In the agronomy studies, it is common to compare new and old technology. You have failed to provide this in the Table 1 and in the text as well.
Response: Thanks. According to your suggestion, we have compared the impact of carbon dots and traditional technologies on crop production in the revised manuscript (in the section of 3.8).

Round 2
Reviewer 1 Report
I think the paper can be published in this form.
The English language was improved.